# Three-Dimensional Simulation Study of the Interactions of Three Successive CMEs during 4–5 November 1998

**Yufen Zhou * and Xueshang Feng**

SIGMA Weather Group, State Key Laboratory for Space Weather, National Space Science Center, Chinese Academy of Sciences, Beijing 100190, China; fengx@spaceweather.ac.cn
* Correspondence: yfzhou@spaceweather.ac.cn

**Abstract:** In this paper, using a 3D magnetohydrodynamics (MHD) numerical simulation, we investigate the propagation and interaction of the three halo CMEs originating from the same active region during 4–5 November 1998 from the Sun to Earth. Firstly, we try to reproduce the observed basic features near Earth by a simple spherical plasmoid model. We find that the first component of the compound stream at 1 AU is associated to the first CME of the three halo CMEs. During the propagation in the interplanetary space, the third CME overtakes the second one. The two CMEs merge to a new, larger entity with complex internal structure. The magnetic field of the first CME in the three successive CMEs event is compressed by the following complex ejecta. The interaction between the second and third CME results in the deceleration of the third CME and the enhancement of the density, total magnetic field and south component of the magnetic field. In addition we study the contribution of a single CME to the final simulation results, as well as the effect of the CME–CME interactions on the propagation of an isolated CME and multiple CMEs. This is achieved by analysing a single CME with or without the presence of the preceding CMEs. Our results show that the CME moves faster in a less dense, faster medium generated by the interaction of the preceding CME with the ambient medium. In addition, we show that the CME–CME interactions can greatly alter the kinematics and magnetic structures of the individual events.

**Keywords:** MHD numerical simulation; CME propagation; CME–CME interaction

## 1. Introduction

Coronal Mass Ejections (CMEs) are powerful solar eruptions that release huge amount of mass into the interplanetary medium (IPM). Their masses can be as large as $10^{15}$–$10^{16}$ g moving outwards at speeds ranging from a few hundred to thousands of kilometers per second [1]. Near solar maximum, CMEs occur at a rate of 3.5 events per day [2], and sometimes several CMEs originate from the same solar region within a relatively short interval [3]. Therefore, they may interact with each other during the propagation from the Sun to 1 AU near solar maximum.

By analyzing in situ observations of CMEs by the Pioneer 9 spacecraft, Intriligator [4] reported the possibility of CME–CME interaction. Moreover, Burlaga et al. [5] presented an additional case of CME–CME interaction in the heliosphere using in situ observations of the twin Helios spacecraft. Using the wide field-of-view (FOV) coronagraphic observations of large angle spectrometric coronagraph (LASCO; [6]) on board the Solar and Heliospheric Observatory (SOHO) and long-wavelength radio observations, Gopalswamy et al. [7] provided the first evidence for CME–CME interaction. In particular, after the launch of the Solar Terrestrial Relations Observatory (STEREO; [8]), the kinematic evolutions of CMEs have been widely studied by using the large field view observations from the Heliospheric Imager (HI) on board the STEREO, e.g., [9–16].

It is known that the kinematics and magnetic field structures of CMEs can be significantly altered during interactions with other CMEs in interplanetary space. The evolutions of CME–CME interaction have been studied by statistical analyses [17–21], as well as analytical study [22,23]. Wang et al. [22] proposed a simple theoretical model to explain how the shock compression of the preexisting southward directed magnetic field may enhance a geomagnetic disturbance. The interaction between multiple CMEs can result in an extended period of an enhanced southward magnetic field, which can cause intense geomagnetic storms [24,25]. By investigating the major geomagnetic storms (Dst $\leq$ $-100$ nT) during 1996–2005, Zhang et al. [19] found that one third (24; 31%) of the 77 CME-driven storm events are associated with a complex solar wind flow produced by multiple interacting ICMEs arising from multiple halo CMEs launched from the Sun in a short period. By analyzing the eight great magnetic storms (Dst $\leq$ $-200$ nT) in solar maximum (2000–2001), Xue et al. [18] found that half of all great storms were related to successive halo CMEs, most of which originated from the same active region. The interactions between successive halo CMEs usually can lead to greater geoeffectiveness by enhancing the southward field Bz interval either in the sheath region or within magnetic clouds (MCs). Temmer et al. [12] studied the interaction of two successive CMEs during the 1 August 2010 events. They found that CME1 represented a magnetohydrodynamic obstacle for CME2, and the strong deceleration of CME2 was explained by interaction with the slower CME1. Lugaz et al. [9] analyzed two interacting CMEs of 23–24 May 2010, and found the deflection of the first CME about 10° toward the Sun–Earth line after the collision of two CMEs. These studies show that the space weather effect of CMEs can be greatly affected during the CME–CME interaction.

In situ measurements outside of Earth's direct vicinity have been limited to planetary missions, which, generally, cannot easily be combined with measurements at 1 AU to study the interaction of successive CMEs and the formation of complex ejecta. 2D and 3D numerical simulation is powerful tool which can give us deeper insight and help us to better understand the evolution of CME–CME interaction in IP space, e.g., [26–33]. Lugaz et al. [27] investigated the complex interaction of two CMEs propagating in the same direction from Sun to Earth by using 3D MHD model. They explained why complex ejecta resulting from interacting CMEs appear to have a homogenized speed [24,34]. Shen et al. [31] numerically studied the event during 2–8 November 2008 through 3D MHD simulations. Results showed that the collision led to extra kinetic energy gain by 3–4% of the initial kinetic energy of the two CMEs. It proved that the collision of CMEs could be superelastic [35]. Lugaz et al. [28] combined numerical simulations of the interaction of two CMEs with different orientations and the analysis of in situ measurements on 19–22 March 2001, and found that such events might result in intense, long-duration geomagnetic storms. Scolini et al. [33] analyzed three successive CMEs that erupted from the Sun during 4–6 September 2017 by using the EUHFORIA model [36], investigating the role of CME–CME interactions as a source of the associated intense geomagnetic storm (Dst$_{min}$ = $-142$ nT on September 7). These numerical studies enhance our understanding about how CME–CME interactions alter their trajectories, morphologies, kinematics, and magnetic structures. However, few of these studies focus on the contribution of a single CME in the CME–CME interaction for real multiple interacting CMEs event by using 3D MHD simulation.

In this study, we select three Earth-directed successive CMEs launched from NOAA Active Region (AR) 8375 during 4–5 November 1998, and carry out the 3D numerical study of their propagation and interaction. The successive CMEs with a similar direction provide a good example of potential interactions of CMEs. Here, we analyze the contribution of single CME to the final simulation results and the effect of the CME–CME interactions on the propagation of a single CME, as well as multiple CMEs. The organization of the paper is as follows. The observations of the 4–5 November 1998 CMEs are given in Section 2, followed by a brief description of the solar wind and CME model in Section 3. The results of the 3D simulation and discussion are presented in Section 4, followed by a summary given in Section 5.

## 2. Observational Properties of the 4–5 November 1998 CMEs

Three successive Earth-directed CMEs were observed by LASCO/SOHO in AR 8375 on 4–5 November 1998, near the beginning of the 23rd solar cycle. The first event was a halo CME that was seen on 4 November 1998, at 0418 UT in C2, in the north, west, and south quadrants. The projected plane-of-sky speed was about 270 km s$^{-1}$ at position angle (PA) 0 (North Pole). Associated with the halo CME, a C5.2 flare began at 0310 UT on 4 November. The second event was a halo CME on 5 November 1998. The event was first visible on at 0241 UT in C2 as a bright mound of emission to the south of the occulting disk. The speed in the plane of the sky was about 380 km s$^{-1}$ at PA 165. The third halo CME occurred late on 5 November, following the M9 flare at about 1945 UT. The event was first visible in C2 at 2044 UT in the northwest quadrant and later developed into a full halo with the projected speed in C3 about 1000 km s$^{-1}$ at PA 310.

The plasma and magnetic field data observed by Wind spacecraft corresponding to the solar events are shown by dotted lines in Figure 1 for the interval 7–10 November 1998. The speed profile in a three day period shows two speed maxima, each preceded by a shock. The first shock reached 1 AU at 07:36 UT on 7 November (vertical solid line), and the second one reached 1 AU at 04:41 UT on 8 November (vertical dashed line). The time delay between the observation of the first halo CME on 4 November and the arrival of the ejecta on 7 November at Wind is about 77 h, giving a mean transit speed of ∼539 km s$^{-1}$, consistent with the observed maximum speed 530 km s$^{-1}$ at 1 AU. The second component passes the spacecraft for two days (8 and 9 November), and has a complicated structure. It probably consists of at least two parts, corresponding to the two halo CMEs observed on 5 November, which interacted and merged into a single component of a compound stream. The more corresponding observations of three successive CMEs have been provided by prior studies [34,37]. For the concrete data of the flares event, one can refer to http://wso.stanford.edu/ (accessed on 9 November 2021).

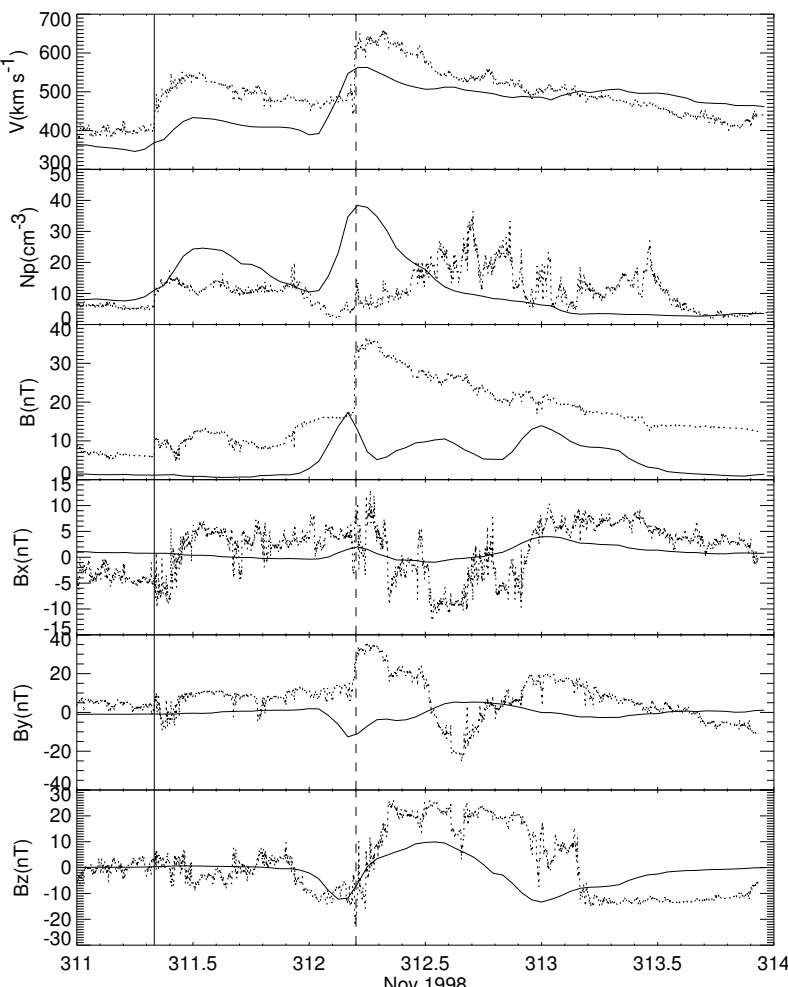

**Figure 1.** Comparison between the in situ data obtained by the Wind spacecraft and the simulation during the 7–10 November 1998 event. From top to bottom, shown are flow velocity, number density, magnetic field, and three components of magnetic field in geocentric solar ecliptic (GSE) coordinates. The two vertical lines indicates the time of the shocks. The simulated results at Earth are shown by the solid lines. The Wind observations are shown by the dotted lines.

## 3. Numerical Model

The numerical technique used in this paper is 3D SIP-CESE MHD model [38,39]. This model can reproduce the solar wind background by means of an artificial heating/acceleration source term in the energy equation, which is given by:

$$S_E = Q_0 \cdot \frac{1}{f_s} \exp(-r/L_Q),\tag{1}$$

where the constant value $Q_0 \cdot \frac{1}{f_s}$ and $L_Q$ are the intensity and decay length of heating, respectively. The constant value of $Q_0$ is set to be $0.6 \times 10^{-6}$ Jm$^{-3}$s$^{-1}$, and $L_Q$ is set to be $0.9\ R_s$. The expansion factor, $f_s = (\frac{R_s}{R_{ss}})^2 \frac{B_{R_s}}{B_{R_{ss}}}$, is determined by using potential field source surface (PFSS) model [40,41] with the source surface at $2.5\ R_s$. Here, $R_s$ and $R_{ss}$ are the solar radius and the source surface radius, and $B_{R_s}$ and $B_{R_{ss}}$ are radial magnetic field strength at the solar surface and at $R_{ss}$. The model uses a variable specific heat ratio $\gamma$ [42]:

$$\gamma = \begin{cases} \gamma_0 + (\gamma_1 - \gamma_0) \sin^2(\frac{\pi}{2} \frac{r - r_1}{r_2 - r_1}) & \text{if} \quad r_1 \leq r \leq r_2, \\ \gamma_1 & \text{if} \qquad r > r_2, \end{cases}\tag{2}$$

In this way, $\gamma$ allows a smooth variation from $\gamma_0 = 1.2$ close to the Sun to $\gamma_1 = 1.46$ near 1 AU. Here, the radial distances $r_1$ and $r_2$ are taken to be 1 $R_s$ and 20 $R_s$, respectively.

The temperature and plasma density at the photosphere are assumed to be $1.5 \times 10^6$ K and $1.34 \times 10^{-13}$ kg m$^{-3}$, respectively. The simulation domain is centered at the Sun and extending to 235 $R_s$. Here, the spatial resolution in the radial direction gradually varies from 0.05 $R_s$ at the inner boundary on the solar surface to 1.0 $R_s$ near 1 AU.

By using the observed photospheric magnetic field data from the photospheric magnetogram and Parker's 1D solar wind solution [43] as the initial values, the quasi-steady solar wind is computed. Figure 2 shows the three-dimensional coronal field and heliospheric current sheet (HCS) structure. It can easily be seen that the magnetic field and radial speed possess a typical characteristic of the rising phase of a solar cycle. The coronal magnetic field shows a mixed dipole/quadrupole configuration with large tilt angle of the dipole. The latitudinal extent of the HCS expands to higher latitudes. Figure 3a shows the steady-state solution in the solar equatorial plane. It can be seen that the interplanetary magnetic field (IMF) lines are stretched by the solar wind outward into Archimedean spirals due to the solar rotation and the IMF freezing-in effect. The magnetic field generates the sector structure due to the different IMF directions in the solar equatorial plane. Figure 3b shows the information on how the solar wind speed is structured in the Sun–Earth meridional plane.

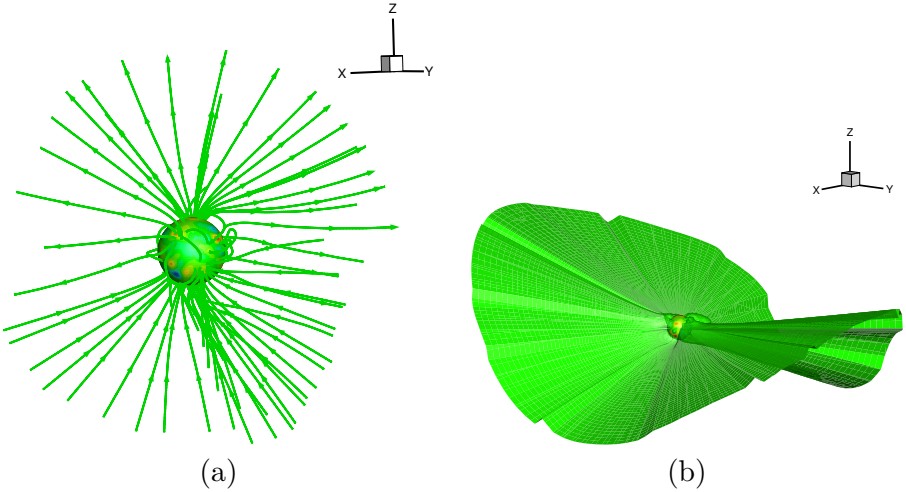

(a)        (b)

**Figure 2.** (**a**) Three-dimensional steady state magnetic field structure and (**b**) steady state current sheet in the solar corona domain.

In the following, the modeled CME is described briefly. A high speed, high density and high pressure spherical plasmoid is superposed on the ambient steady state solar wind to generate the CME. The other parameters of the initial perturbation, including the density, temperature and radial velocity, are defined as follows:

$$\begin{aligned}
\rho &= \rho_0 + \rho_{\max}(1 - a^2/a_{\mathrm{cme}}^2) \\
T &= T_0 + T_{\max}(1 - a^2/a_{\mathrm{cme}}^2) \\
v_r &= v_{r0} + v_{\max}(1 - a^2/a_{\mathrm{cme}}^2)
\end{aligned} \tag{3}$$

where $a_{\mathrm{cme}}$ and $a$ denote the radius of the plasmoid and the distance from its centre, respectively. $\rho_0$, $v_{r0}$ and $T_0$ are the density, radial velocity and temperature of the ambient solar wind, respectively. $\rho_{\max}$, $v_{\max}$ and $T_{\max}$ are the maximum density, radial velocity and temperature superimposed on top of the ambient solar wind, respectively.

In the spherical plasmoid, the initial magnetic field is taken to be the following form [39] in local spherical coordinates $(r^{\ell}, \theta^{\ell}, \phi^{\ell})$:

$$
\begin{aligned}
B_{r^{\ell}} &= (2B_0/\alpha r^{\ell})j_1(\alpha r^{\ell})\cos\theta^{\ell} \\
B_{\theta^{\ell}} &= -(B_0/\alpha r^{\ell})[\sin(\alpha r^{\ell}) - j_1(\alpha r^{\ell})]\sin\theta^{\ell} \\
B_{\phi^{\ell}} &= \pm B_0 j_1(\alpha r^{\ell})\sin\theta^{\ell}
\end{aligned}
\tag{4}
$$

where $\alpha = 4.493409458a_{\mathrm{cme}}^{-1}$ is derived from the force-free condition of $\nabla \times \mathbf{B} = \alpha\mathbf{B}$ with the boundary condition of $B_{r^{\ell}} = 0$ at $r^{\ell} = a_{\mathrm{cme}}$. $B_0$ is a constant magnetic field strength superimposed on the ambient solar wind, and $j_1(x) = x^{-2}\sin x - x^{-1}\cos x$. The parameter $\alpha$ becomes negative for left-handed polarity. The reader refer to Feng [44] and references therein for more details of simulating the ambient solar wind and solar eruptions.

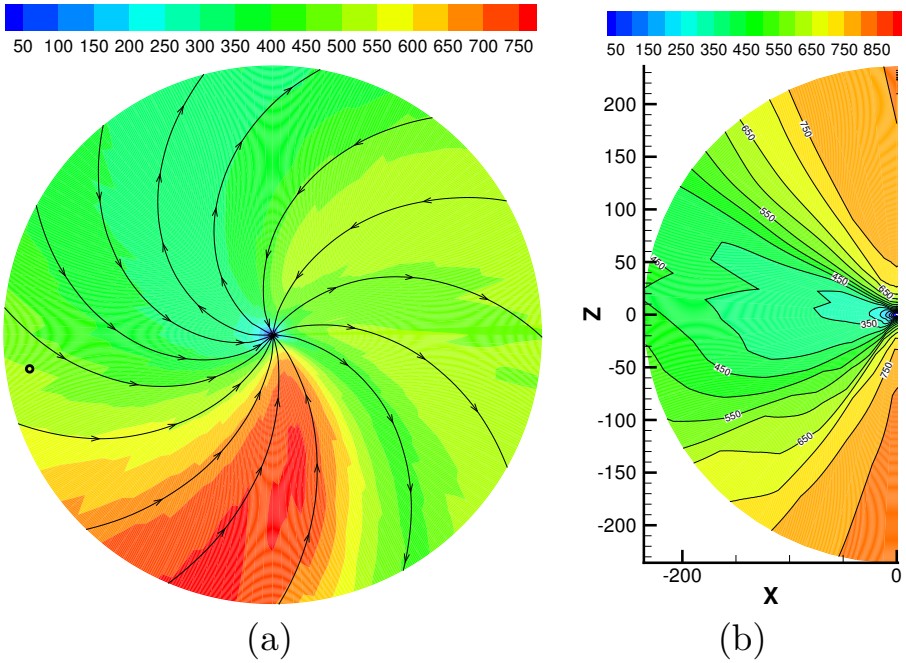

**Figure 3.** (**a**) Steady state solution in the solar equatorial plane. The black streamlines denote the magnetic field lines. The black circle represents Earth. (**b**) Steady state solution in the Sun–Earth meridional plane. The color contours represent the radial speed (km s$^{-1}$).

The initial parameters corresponding to the three successive CMEs are shown in Table 1. The three successive CMEs are launched from N17E01, N19W11 and N22W18 to conform to the locations of the flares/CMEs, respectively. The initial speeds of the three CMEs come from the catalog by Michalek et al. [37], determined by the observation data of LASCO/C2 combined with CME's "Cone Model". According to their occurrence time, the three successive CMEs are added at t = 0 h, 23.5 h, and 40.5 h, respectively. That is, the launch time of each CME corresponds to the flare onset time on the Sun. The three CMEs are launched at 2.0 $R_s$ with the radiuses of 0.8 $R_s$. In the simulation, we empirically choose the other parameters of the CME model to reasonably match the simulated properties of the ICMEs with in situ measurements at 1 AU. Other parameters involved in the perturbation are given in Table 1 after tuning, since they are not available directly from observations. The corresponding energies added to the corona are also given in Table 1.

**Table 1.** Perturbation parameters (velocity unit: km s$^{-1}$, density unit: $10^7$ cm$^{-3}$, temperature unit: $10^6$ K) used in simulating the three successive CMEs and their corresponding energies (E represents energy with unit: $10^{31}$ ergs).

| | $V_{max}$ | $\rho_{max}$ | $T_{max}$ | $B_0$ | Magnetic E | Kinetic E | Thermal E | Total E |
|---|---|---|---|---|---|---|---|---|
| CME1 | 270 | 0.73 | 2.47 | 3.0 | 2.46 | 0.026 | 0.297 | 2.78 |
| CME2 | 380 | 0.73 | 2.44 | 3.0 | 2.42 | 0.050 | 0.297 | 2.77 |
| CME3 | 1000 | 1.41 | 2.38 | 6.0 | 9.74 | 0.507 | 1.483 | 11.73 |

## 4. Simulation Results and Discussion

We first perform a simulation of the propagation and interaction of three successive CMEs on 4–5 November 1998 (labeled as CME123) using the 3D SIP-CESE MHD model. A three-dimensional view 50 h after the eruption of the first CME is shown in Figure 4, where the lines represent the magnetic field. Here the fronts of the three CMEs are shown as the isosurfaces of $\bar{\rho} = 1.5\rho_{wind}$ and false color representations of velocity magnitude cover the isosurfaces, where $\rho_{wind}$ is the density of the background solar wind. When the front of the first CME reaches 147 $R_s$, the second one and the third one reach 83 $R_s$ and 51 $R_s$, respectively. Figure 1 displays the comparison of our simulated results at Earth and the Wind in situ observations. According to the solar wind data observed by Wind, the first shock arrives at 1 AU with density 16 cm$^{-3}$ and velocity 530 km s$^{-1}$, and the second component with velocity 635 km s$^{-1}$ and maximum density 36 cm$^{-3}$. The simulated parameters show that the first component has velocity 434 km s$^{-1}$, 96 km s$^{-1}$ slower than the observed and density 23 cm$^{-3}$, 8 cm$^{-3}$ greater than the observed data, respectively. The simulated velocity and density of the second component are in good agreement with the observed values. The simulated results can successfully reproduce two speed maxima corresponding to two components consistent with the observation. The arrival time of the first shock is almost consistent with that of the interplanetary shock observed by Wind spacecraft. The computed second shock arrives at Earth 3 h earlier than the observed. Here, we use the value of the relative density $(\rho - \rho_{wind})/\rho_{wind}$ being 0.5 as the criterion to identify the position of the shock front, where $\rho$ is the total density and $\rho_{wind}$ is the density of the background solar wind. Because the numerical study is computed by only two-order scheme, and the spatial resolution is 1.0 Rs near 1 AU, the results can't achieve high shock resolution as expected with the high resolution shock capturing schemes. In future, we can efficiently capture strong gradients by constructing high-order accurate shock capturing scheme or using adaptive mesh refinement.

The computed magnetic field shows the increase in the total magnetic field magnitude, and in particular the magnetic field component Bz, similar to that observed by Wind spacecraft. Because the simulated magnetic field is small, the computed magnetic field in Figure 1 is multiplied by 2 in order to compare our simulated results at Earth and the Wind in situ observations. This may be due to the small background magnetic field, which is caused by the imperfectness of potential field source surface model and the coarse grid resolution in IP space. There are some differences between the simulated x and y component of the magnetic field and the observed data. Despite its shortcomings, our simulation gives the main features of the in situ data at 1 AU. Magnetic field vectors at 1 AU are affected by the magnetic parameter of the model (e.g., amplitude, longitudinal and latitudinal sizes, and tilt angle of the spherical model). These parameters are given ad hoc way to reconstruct the Wind data. Therefore, there is no guarantee that these initial parameters are correct, meaning that other CME parameters can explain the in situ data. However, this study focuses only on the difference between single CME and the CME–CME interactions. The self-consistent reconstruction of Bz is beyond the scope of this study. Therefore, this uncertainty of the CME parameters does not change the result of this study.

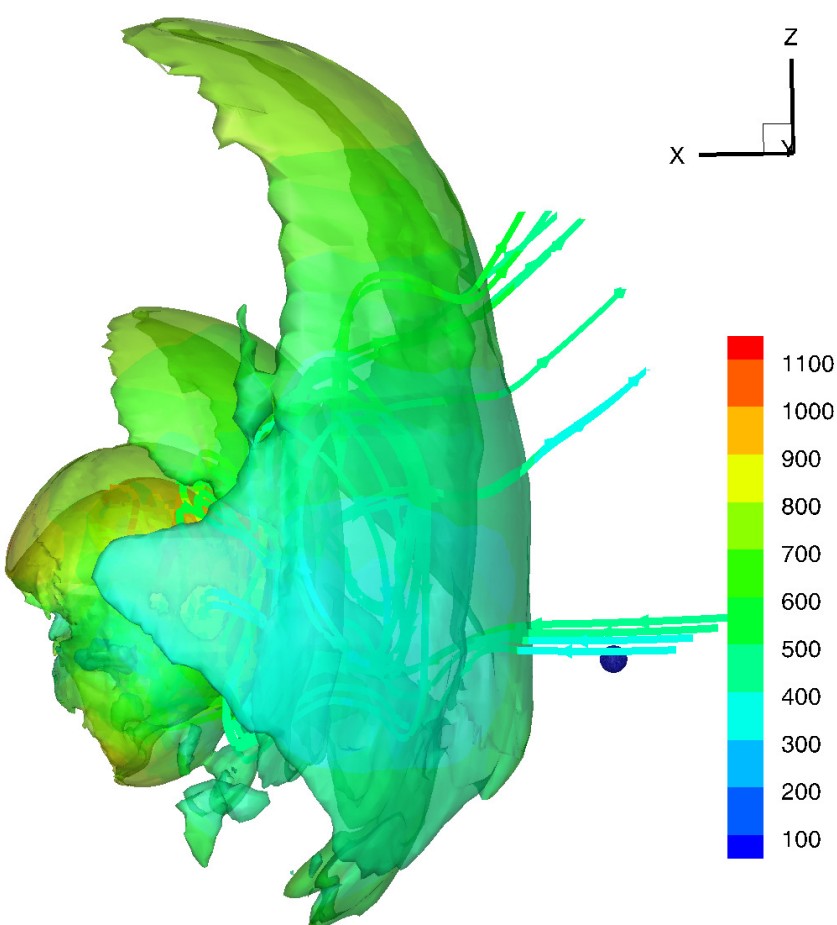

**Figure 4.** A 3D representation of the CME is shown at 50 h after the initiation of the first CME. The color code represents the velocity magnitude. Three-dimensional colored isosurfaces of $\overline{\rho} = 1.5\rho_{wind}$ are drawn. The blue sphere shows the position of the Earth with a radius of 5 $R_s$.

In order to better isolate the contribution of single CME to the final simulation results and the effect of the CME–CME interactions on the propagation of the isolated CME, we simulate an isolated event by simply propagating one of the three CMEs without the other eruptions. Hereafter, we use the same background solar wind for all the cases simulated and keep the CME parameters in all the cases exactly the same as those in CME123. Therefore, the choice of CME parameters does not affect the following discussion. Here, we use CME1, CME2 and CME3 for the first, second and third isolated CME. Then, we consecutively add the CMEs in the simulations to see how the simulation results change.

We first perform a simulation with CME1 alone, CME2 alone, and two successive CMEs which are introduced with a separation time of 23.5 h (labeled as CME12). Figure 5 shows a comparison of the simulation results along the direction of Earth for three different cases: CME1, CME2 and CME12. In order to compare the results, we shift the result of CME2 alone by 23.5 h behind to the time axis of simulated profiles. Seen from the result of Figure 5, the second CME of CME12 propagates faster than isolated CME2, and its density is smaller than that of the isolated CME2. This is because the interaction of the first CME with the ambient solar wind removes some of the background's mass. Thus the second CME propagates into a less dense, faster background solar wind. This analysis was also mentioned by Lugaz et al. [27]. Ultimately, the transit time of the shock from Sun to Earth is 78 h and 81 h for the second CME of CME12 and CME2 alone, respectively, which demonstrates that the interaction of the first CME and the ambient solar wind can obviously influence the arrival time of the shock at Earth for the second CME of two successive CMEs. From Figure 5c, we can find that the density and velocity of the first CME of CME12 are the same as those of CME2 alone. However, the magnetic field of the first CME is affected by

the second one in CME12. It can be seen that the southward magnetic field of the first CME is compressed by the second one, resulting in small enhancement of the amplitudes of the magnetic field and southward magnetic field. This may be due to the shock front driven by the following CME propagating into the preceding one, compressing the magnetic field inside the ICME, and thus enhancing its geoeffectiveness, as found in previous studies, e.g., [20–22].

We next perform a simulation by adding CME3 17 h after CME2 (labeled as CME23). Since the second CME of CME23 move faster than the first one, the two CMEs get closer and closer as shown in the four panels of Figure 6. There was a clear and visible interaction between the two CMEs. This result can also be seen from Figure 7 that shows a height/time map of the relative density distribution obtained along the Sun–Earth direction from time t = 0 h to 70 h expressed in solar radii from 1 to 215 $R_s$. We see that two different trajectories, which correspond to the two CMEs, form. The first and second trajectory curves in Figure 7 correspond to the first CME and the second one of CME23, respectively. The second trajectory shows the position of the second CME, respectively. At about 32.5 h, the front edge of the second CME overtakes the trailing edge of the first CME at 77 $R_s$. The merging of the trajectory curves indicates the interaction of the two CMEs, i.e. merging of the two CMEs into a new, larger entity with complex internal structure. Before collision, the speed of the first CME front is 415 km s$^{-1}$ at 98 $R_s$. The second CME front is found at 77 $R_s$ with the speed of approximatively 650 km s$^{-1}$. After the collision, the average speed of the complex ejecta decelerates to about 550 km s$^{-1}$. From this, we attribute the deceleration to the interaction of the two CMEs, which is consistent with Temmer et al. [12].

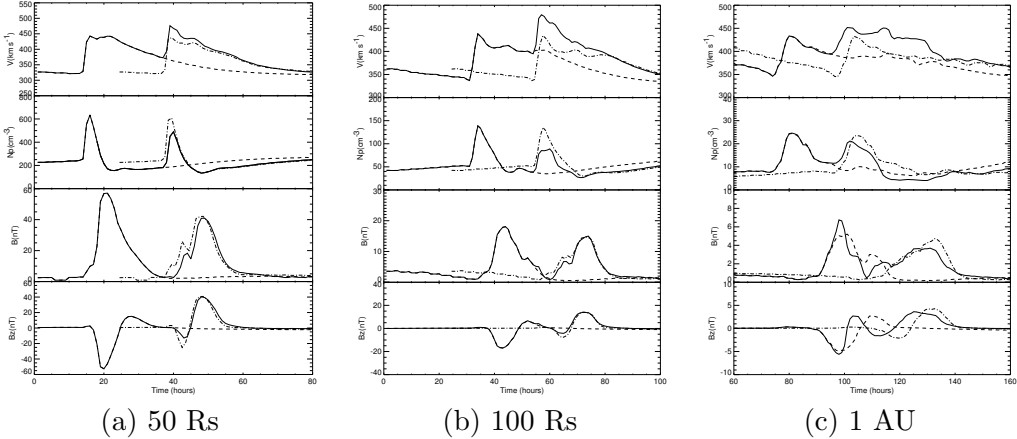

**Figure 5.** Profiles of velocity, density, magnetic field and south component of magnetic field as function of time at different heliospheric distances: (**a**) 50 $R_s$ (**b**) 100 $R_s$ and (**c**) 1 AU for CME1 (dashed line), CME2 (dash-dotted line) and CME12 (solid line).

We next perform a simulation to analyze the propagation of CME3 with or without the presence of CME2. At 50 $R_s$, the density of the second CME in the CME23 simulation is smaller than that of the isolated CME3, related to the removal of the background's mass via the interaction of a first CME with the ambient solar wind. However, the density of complex ejecta increases at 100 $R_s$ due to the collision of two successive CMEs. These results can be seen from Figure 8. Ultimately, the density of CME23 is 11 cm$^{-3}$ higher than that of isolated CME3 at 1AU. After the interaction, the merged entity propagates as a larger structure in an ambient medium. This merging leads to more complex magnetic field structures, making the interpretation of merged CMEs more difficult at 1AU.

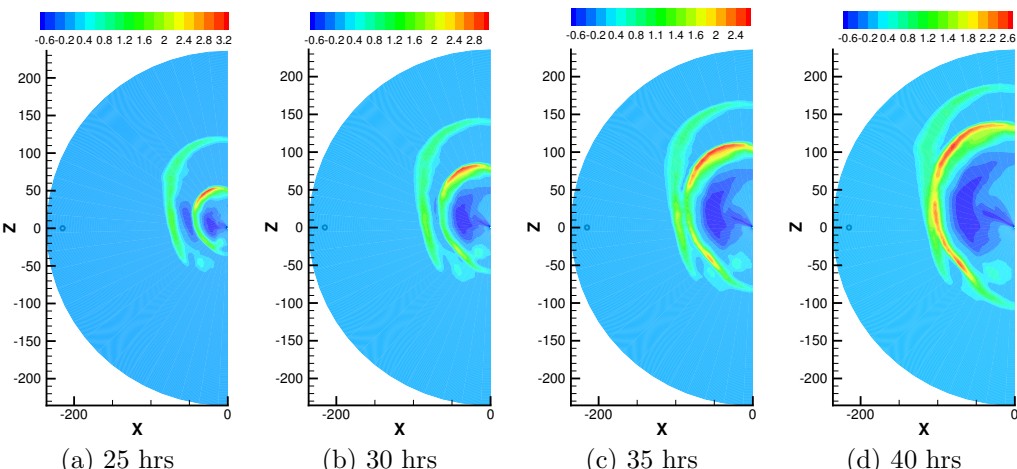

**Figure 6.** The contour plots of the relative density $(\rho - \rho_{wind})/\rho_{wind}$ distribution on the solar-terrestrial meridional plane after (**a**) 25 h (**b**) 30 h (**c**) 35 h and (**d**) 40 h of the first CME of CME23, where $\rho$ is the total density and $\rho_{wind}$ is the density of the background solar wind.

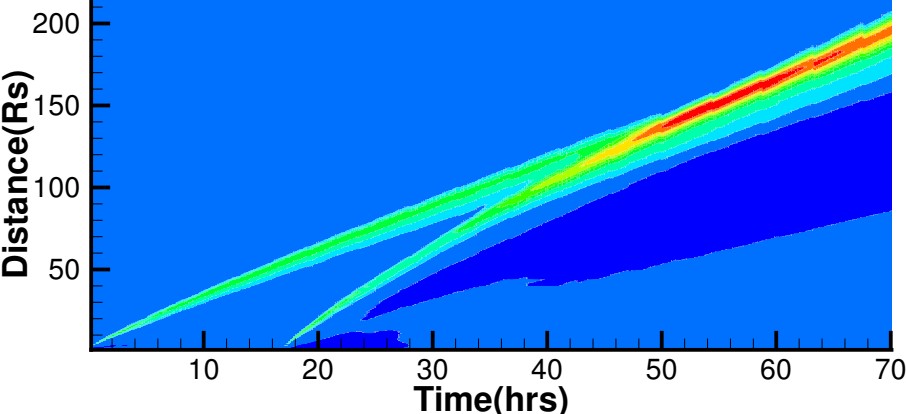

**Figure 7.** The height/time map of the relative density distribution obtained along the Sun–Earth direction during 0–70 h expressed in solar radii from 1 to 215 $R_s$ for CME23.

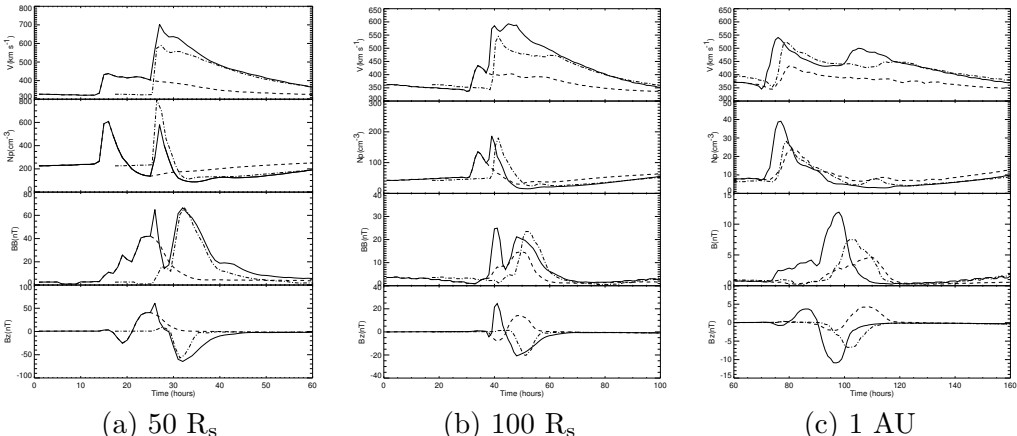

**Figure 8.** Profiles of velocity, density, magnetic field and south component of magnetic field as function of time at different heliospheric distances: (**a**) 50 $R_s$ (**b**) 100 $R_s$ and (**c**) 1 AU for CME2 (dashed line), CME3 (dash-dotted line) and CME23 (solid line).

It is also interesting to see how the CME–CME interaction changes the geo-effectiveness of these events at Earth. To predict the intensity of a magnetic storm, the most important

parameter is the z component of the magnetic field. From Figure 8a, we find that the inter-action between the magnetic structures of the two CMEs starts earlier than their leading edges merging in the heliospheric images. The flux rope of preceding CME is compressed by the following one, and the compression enhances the total magnetic field and more notably, its southward oriented component. This can be seen from the result in Figure 8b,c. Several studies have established that CME–CME interactions are likely to increase the impact on Earth of individual CMEs, e.g., [18,45]. These results show that CME–CME interactions can greatly alter the morphology, kinematics, and magnetic structures of the individual events [13].

Finally, we make a comparison of the simulation results among four different cases: CME123, CME1 only, CME2 only and CME3 only. Figure 9 shows the profiles of the parameters as function of time along Earth direction at different heliospheric distances for CME1 (red line), CME2 (blue line) CME3 (green line) and CME123 (black line). From the figure, we can see that the density of the second CME and the third CME of CME123 is smaller than that of individual CME2 and individual CME3 at 50 $R_s$. In addition, the density difference between the third CME of CME123 and CME3 alone is greater than that between the second CME of CME123 and CME2 alone. However, the density enhances during the interaction of the third CME and the second CME, which can be seen from Figure 9b. Ultimately, the density of the second compound of CME123 is higher than that of isolated CME3 at 1 AU. The clear and visible interaction between the second CME and the third CME can also be found in Figures 10 and 11. From Figure 11, three different trajectories, which correspond to the three CMEs, can be well seen. The first CME creates the first track. The second and third trajectories show the position of the second CME and the third one, respectively. 59 h after the launch of the first CME, the leading edge of the third CME overtakes the trailing edge of the second one. At the beginning of the collision, the leading edge of the second CME is around 95 $R_s$, whereas the leading edge of the complex ejecta after the collision phase is around 135 $R_s$. During the interaction, the speed of the third CME decelerates from 700 km s$^{-1}$ to 602 km s$^{-1}$, the latter corresponding to the flow speed of the complex ejecta. The third CME interacts with the second one 35.5 h after the launch of the second one, which is later than the interaction time of these two CMEs in the CME23 simulation. This is because the second CME in the CME123 simulation propagates faster compared to the first CME in the CME23 simulation due to preconditioning of the interplanetary space by the first CME in the CME123.

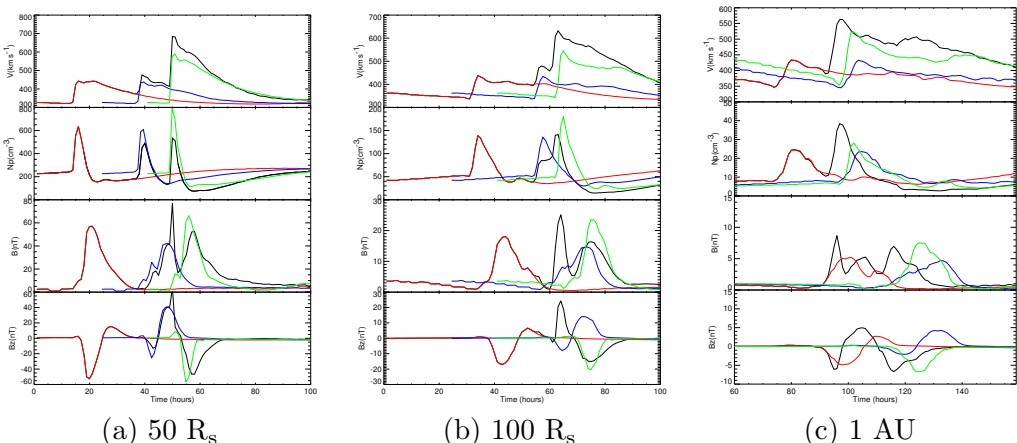

**Figure 9.** Profiles of velocity, density, magnetic field and south component of magnetic field as function of time at different heliospheric distances: (**a**) 50 $R_s$ (**b**) 100 $R_s$ and (**c**) 1 AU for CME1 (red), CME2 (blue), CME3 (green) and CME123 (black).

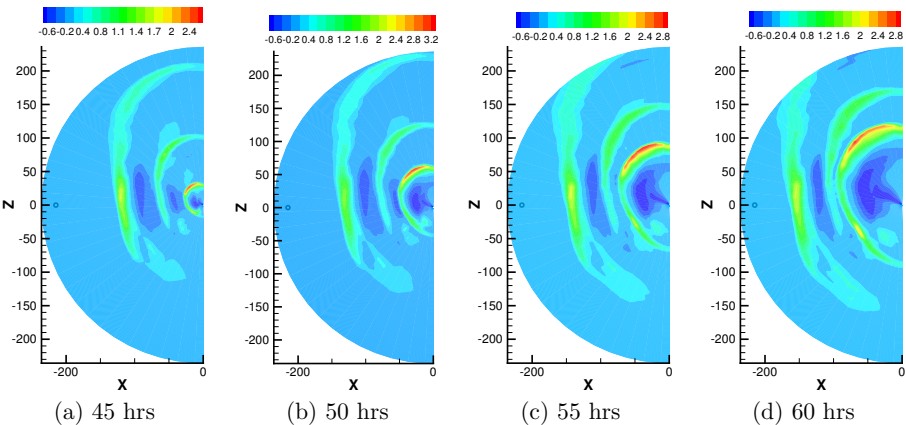

**Figure 10.** The contour plots of the relative density distribution on the solar-terrestrial meridional plane after (**a**) 45 h (**b**) 50 h (**c**) 55 h and (**d**) 60 h.

Ultimately, the first CME forms the first component of the compound stream in Figure 1. During the propagation in the interplanetary space, the two following CMEs merge to form new complex ejecta, the second component of the compound stream. This result is consistent with the analysis of Burlaga et al. [34]. The identity of the individual CMEs cannot be well discerned in the complex ejecta at 1 AU. The second component may interact with the first one during their propagation outside 1 AU owing to the higher velocity. Simulations of the magnetic field show that the total magnetic field and especially the southward component of the magnetic field are enhanced during the interaction between the magnetic structures of the CMEs, which can lead to greater geoeffectiveness. The magnetic field of the first compound of CME123 is compressed by the second compound.

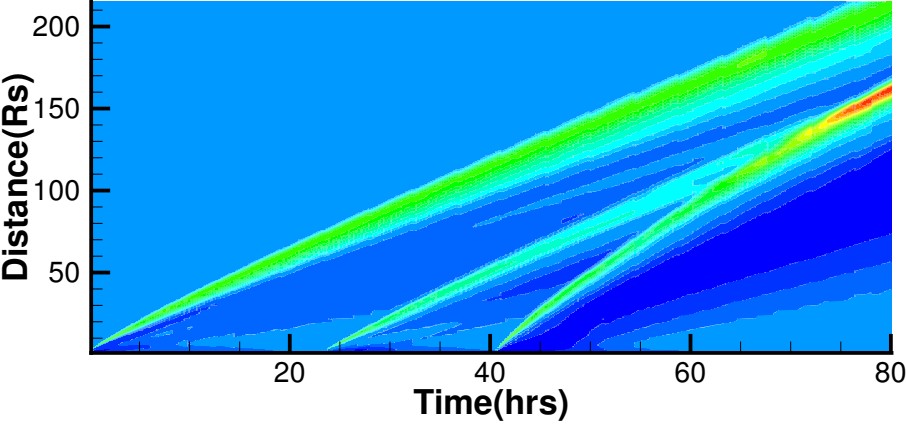

**Figure 11.** The height/time map of the relative density distribution obtained at the Earth-direction during 0–80 h expressed in solar radii from 1 to 215 $R_s$ for CME123.

## 5. Summary

In this study, we investigate the evolution and interaction of the three CMEs during 4–5 November 1998 based on the real background solar wind by using a 3D MHD numerical simulation. Multiple CMEs with a similar direction provide a good example of potential interactions of CMEs, which gives us an opportunity to numerically study the evolution of these interactions.

By comparing the simulation results with the Wind-measured data, we find that the MHD results can successfully reproduces many basic features of the in situ measurements: Such as the two large-scale structures, and the variability of the magnetic field. The first component of the compound stream corresponds to the first of the three CMEs. During the propagation in the interplanetary space, the third CME overtakes the second one. Then

the two CMEs interact with each other and merge into a new, larger entity with a complex internal structure.

Then we simulate a single CME with or without the presence of the preceding CMEs, and analyze the effect of a single CME and the CME–CME interactions on their final simulation results. By comparing the results of CME1 alone, CME2 alone and CME12, we find that the second CME in CME12 simulation propagates faster than isolated CME2, and the density of second CME in CME12 simulation is smaller than that of the isolated CME2 due to the preconditioning of the interplanetary space. Ultimately, the arrival time of the second CME in CME12 simulation at Earth is earlier than that of isolated CME2. Thus, the preconditioning affects CME arrival time at Earth. At 1 AU, the shock front driven by the second CME propagates into the first CME, the shock compresses the flux rope of the first CME and thus enhancing its total magnetic field.

The results of CME23 show that the two successive CMEs get closer and closer during their propagations as the following CME is faster than the preceding one. At 32.5 h, the front edge of the second CME overtakes the trailing edge of the first one and the two of them merge. The resulting entity is decelerated and denser compared to CME3 alone. The preceding CME represents a magnetohydrodynamic obstacle for the following CME. The interactions between the two successive CMEs enhance the total magnetic field and more notably its southward component. The result for the second and third CMEs in CME123 simulation is similar to that obtained from CME23 simulation. Our study shows that CME–CME interactions can greatly alter the morphology, kinematics, and magnetic structures of the individual events. The analysis about the effect of the CME–CME interactions on the propagation of a single CME and interacting CMEs improves our understanding about the nature and consequences of CME–CME interactions.

**Author Contributions:** X.F. proposed the problem while Y.Z. did the calculations and finalized the manuscript. All authors have read and agreed to the published version of the manuscript.

**Funding:** This research is jointly supported by the B-type Strategic Priority Research Program of Chinese Academy of Sciences, (Grant No. XDB 41000000), the National Natural Science Foundation of China (Grant Nos. 41861164026, 41874202 and 42030204), and the Specialized Research Fund for State Key Laboratories. The work was carried out on TianHe-1(A) for National Supercomputer Center in Tianjin, China.

**Institutional Review Board Statement:** Not applicable.

**Informed Consent Statement:** Not applicable.

**Data Availability Statement:** The Wilcox Solar Observatory data used in this study were obtained via the Web site http://wso.stanford.edu (accessed on 9 November 2021) at 2009:05:06_17:15:30 PDT courtesy of J.T. Hoeksema. The Wilcox Solar Observatory is currently supported by NASA. We also thank OmniWeb, http://omniweb.gsfc.nasa.gov/ (accessed on 9 November 2021), from which we downloaded the hourly average solar wind data by WIND.

**Conflicts of Interest:** The authors declare no conflict of interest.

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
