# Peer review of "Three-Dimensional Simulation Study of the Interactions of Three Successive CMEs during 4–5 November 1998"

_universe, doi:10.3390/universe7110431_

Round 1

Reviewer 1 Report

This paper discusses the 3D MHD simulation of the evolution and interaction of 3 halo CMEs that originated form the same region on November 4-5, 1998. In the simulation, the third CME overtakes the second one and these two CME merge to a new, complicated structure. This event is analyzed in detail.

The paper is well-written and clear.

I have the following remarks/suggestions:

  1. On page 2 cite the EUHFORIA paper (Pomoell and Poedts, 2018) when mentioning the EUHFORIA model.
  2. The heating term given in Eq. (1) is only depending on the radial coordinate and Br (through the expansion factor) and also the specific heat ratio applied only depends on the radial coordinate. It is somewhat surprising to me that this is sufficient to get a bimodal wind. I wonder how good or bad it is in comparison with other models…
  3. One or more words are missing in the sentence “The result shows that 130 the magnetic generates the sector structure in interplanetary space.” on page 4, which I do not understand. What is “the sector structure”? and is it caused by the magnetic field? How?
  4. The parabolic profiles of the CMEs superposed on the background wind (Eq.(3)) are not smooth, the derivatives are not continuous at the boundary. This must result in numerical noise in the beginning of the simulations. I wonder how substantial this noise is?
  5. The magnetic field of the plasmoids seems to be defined in another ‘local’ spherical coordinate system. I wonder what the relation between rl and a They both indicate the radius of the plasmoid so I think they are the same? Why are there two local spherical coordinate systems used? Is this not confusing?
  6. In fact, the magnetic field specified in Eq.(4) is almost that of a linear force-free spheromak, except for the theta component. Is there a typo and is this the spheromak solution, or is it a different force-free solution? In what way it is different then?
  7. The initial speeds of the CMEs are determined using the cone model. However, Scolini et al. A&A, 626, A122 (2019) show that one needs to adjust the launch speed of a spheromak in order to take into account the additional expansion rate due to the internal magnetic field of the spheromak. Was such an adjustment not necessary here?
  8. Figure 4 shows that the modeled background wind is too slow at Earth (1 AU) by about 40 km/s (10%). When this would be adjusted the fit of the L1 data would be better.
  9. The plots mention x,y,z coordinates and magnetic field components but it is not specified what coordinate system this is and how it is related to the spherical coordinates used before. The V shown in Figure 4 is the radial velocity component I presume?
  10. At the bottom of page 6 a “potential field source surface model” is mentioned that was not mentioned before? Is this for initiating the coronal field?

This paper needs some further clarifications following the suggestions and questions above. The simulations are simple but seem to be able to capture the basic physics. I particularly like the analysis in section 4 where the CMEs are first simulated all together and then taken one by one and in pairs in order to quantify the effect of preceding CMEs on the following ones and on the CME-CME interactions that are observed to occur and the geo-effectiveness of these CMEs and CME interactions. Well done!

Reviewer 2 Report

Referee Report

Manuscript ID#   universe-1421021

Title: Three-dimensional Simulation Study of the Interactions of Three Successive CMEs During November 4-5, 1998

This work studies the effect of interactions between CMEs - background solar wind and CME - CME using 3D MHD simulation, that would be of interest to the readers of this journal. A series of numerical simulations that includes single and multiple CMEs successfully shows the effects of CME-solar wind and CME-CME interactions. The numerical simulation seems to be performed correctly and results are shown appropriate way. There are several comments that could be addressed before the publication.

Major comment

Page 2 line 80

The purpose of this study is not clear. This manuscript just shows a series of CME simulations and mention the properties of the results. Here, please mention more specifically what is the unknown part of the physics, and what topics this study investigates.

Then, in the final section, it would be better to add some sentences to summarize what is investigated to the purpose of this study.

Page 5 line 150;

“In the simulation, we empirically choose the other parameters of the CME model to match the simulated properties of the ICMEs reasonable well with in situ measurements at 1 AU.”

And Page 6 line 178 “The computed magnetic field is multiplied by 2.”

Please explain the choice of CME parameters does not affect the discussion of this study.

It seems that the initial parameters related to the CME magnetic field, i.e., the parameters of the spherical plasmoids are given ad hoc ways to reproduce the 1AU data. Then, this study discusses the simulated |B| and B components at 1AU in the result and discussion session. Therefore, it is natural that the simulated results show similar profile with the Wind data and this simulation does not self-consistently simulate Bz of the CME-CME interactions. In this kind of study, there are many free parameters that affect the in situ characteristics at 1 AU, therefore, the observed time profiles can be explained by other ways. If the spherical plasmoid parameters would be given from independently by other data such as the GCS fitting of LASCO and/or STEREO/CORs/HI, or in situ data taken inner orbit, then the comparison between the simulated and observed data make sense (but it would be difficult for this event).

It seems that the scope of this study is to show the effect of CME-CME and CME-SW interactions (that means how the plasma properties are changed by the interaction), and the self-consistent prediction of Bz is beyond the scope of this study.

Figure 4 and related part.

It would be better to mention more about the resolution of the simulation. The shock structures are resolved enough? In other words, how many grids are allocated to the shocks at 1AU?

Minor comment

Page 2 line 59

“effected” is “affected”?

Page 2 line 70

“earth” should be “Earth”

Page 3 line 104

Figure 4 is mentioned here earlier than Figures 1-3. It should be Figure 1.

Page 3 line 114

The URL is missing.

Page 3 line 115

“~~~the magnetic field strength at the photosphere and at radial distance r = 2.5 Rs.”

I assume that this study used a synoptic magnetogram by WSO and extrapolate the coronal magnetic field using PFSS with r = 2.5 Rs. Please mention it here.

Page 4 line 130

“the magnetic generates” would be “the magnetic field generates”

Figure 2 caption

Please explain “E0”.

Page 6 line 173

“The computed first shock arrives at Earth 3 hours later than the observed. And the arrival time of the second shock is almost consist with that of the interplanetary shock observed by WIND spacecraft.”

It is confusing because, in Figure 4, it seems that the simulated and observed first CME arrived at the same time. It would be better to put computed arrival times as additional vertical line in Figure 4.

Page 8 line 208

“Meanwhile, its amplitude of magnetic field increases.”

It is not clear in Figure 5c.

Page 8 line 215

“the relative density distribution”

Is it a ratio between with/without CME? Or, is it subtraction? Please mention it.

Page 10 line 249

Figure 11 mentions earlier than Figure 10.

Page 11 line 282

“5. Discussion” There are “4. Result and Discussion” session. Therefore, the section 5 should be something else such as summary or conclusion.

All over the manuscript

It is suggested that the English of the manuscript should be edited by a native editor.

Round 2

Reviewer 1 Report

I am very happy with the replies to the points I raised in my previous report and with the corresponding modifications made in the manuscript. I have no further remarks. These are nice results and should be published.

Author Response

Dear Reviewer

Thank you very much.

Best Regards

Reviewer 2 Report

Referee Report

Manuscript ID#   universe-1421021 R1

Title: Three-dimensional Simulation Study of the Interactions of Three Successive CMEs During November 4-5, 1998

The manuscript has been revised partially appropriately. However, some replies are not enough to publication. Therefore, there are still some comments to improve the manuscript.

Replay to the Comment:

“ANS: Because the simulated magnetic field is small, only the computed magnetic field in figure 4 is multiplied by 2 in order to compare our simulated results at Earth and the WIND in situ observations.

In this study, we use the same background solar wind for all the cases simulated and keep the CME parameters in all the cases exactly the same as those in CME123. And these cases are launched at the same latitude and longitude as those from observations. So the choice of CME parameters does not affect the discussion of this study.”

Please mention this reply in the manuscript.

Replay to the Comment:

“However, in this study, two key parameters, their initial propagation directions and velocities, are chosen to be the same as those derived from observations [Burlaga et al. 2002, Michalek et al. 2003].”

It is not enough to answer the comment. Magnetic field vectors at 1 AU should be affected by the magnetic parameter of the model (e.g., amplitude, longitudinal and latitudinal sizes, and tilt angle of the spherical model). And these parameters are given ad hoc way to reconstruct the Wind data in this study. Therefore, there is no guarantee that these initial parameters are correct. This means that other CME parameters can explain the in situ data. For example, it is correct that the Bz enhancement that discussed in Figure 5/8 are generated by the CME-CME interactions in this simulation, but other possibilities can explain the real in situ data. However, this study focuses only on the difference between single CME and the CME-CME interactions. And the self-consistent reconstruction of Bz is beyond the scope of this study. Therefore, this uncertainty of the CME parameters does not change the result of this study. If the authors agree with this comment, it is suggested that above discussions are mentioned somewhere in the manuscript.

Reply to comment:

“ANS: Because the numerical study in this manuscript is computed by only two-order scheme, and the spatial resolution is 1.0Rs near 1AU, the results can’t achieve high shock resolution as expected with the high resolution shock capturing schemes. Ultimately, the shock structures in this manuscript are not resolved enough. In future, we can efficiently capture strong gradients by constructing high-order accurate shock capturing scheme or using adaptive mesh refinement.”

First, please mention this reply in the manuscript. Then, please explain why this limited resolution does not affect the result. For example, this manuscript discusses “faintly increase” in Page 9 line 233. If the shock structure is not resolved sufficiently, the peak intensity should have an error.
